# The Reception of Bantu Divination in Modern South Africa: African Traditional Worldview in Interaction with European Thought

Ullrich Relebogilwe Kleinhempel

Department of Philosophy, Faculty of Humanities, University of Zululand, KwaDlangezwa 3886, South Africa; u.kleinhempel@gmail.com

**Abstract:** Bantu African divination is firmly established in South Africa in the context of modernity and is protected, endorsed and regulated by law. It is received in the therapeutic field. Important explorations were performed in the early 20th century by psychiatrists and psychoanalysts of Jungian orientation. Their cultural, philosophical, spiritual, and academic backgrounds are relevant to this reception. Jungian thought, Spiritual Spiritism, and traditions of European philosophy of divination resonated with the experience, observation, and understanding of Bantu divination. ('Bantu' designates the cultural and linguistic realm from Cameroon and Kenya southwards). Religious-philosophical traditions, as well as the conceptualisations of 'divination' by Plutarch and Iamblichus, are preserved. The reception and appreciation of Bantu divination in South Africa emerged from it, and resonated with these European traditions of religious-philosophical thought. Out of this development a distinct 'South African modernity' emerges. A parallel reception process developed in Brazil, in the belief systems of Umbanda and Kardecism. These developments are illustrated at present in the literatures of South Africa and Brazil, specifically in Afrikaans literature, black South African poetry and its poetics, and Magic Realism in Brazilian literature. Lastly, a perspective is offered of modernity's reception by black scholars and diviners, continually interacting with Jungian psychoanalysis.

**Keywords:** divination; African traditional world view; psychoanalysis; analytical psychology; intercultural studies; postsecularism; multiple modernities; syncretism



## 1. Introduction

This paper is dedicated to the reception of Bantu divination in modern South Africa, with a view to a similar development in Brazil. Divination is part of religion, especially in southern African traditions, but it is not religious per se. It connects the fields of psychotherapy and religion, extending into both, but also links to realms such as the 'prognostic' and 'magic'. The concept of 'divination' was developed extensively in European philosophical tradition by Iamblichus of Chalcis. In his work *De Mysteriis* (written circa 300 C.E.), he reflected on ancient traditions and forms of divination, and integrated them into the frame of Neoplatonic philosophy. He reflected on the relation of divination with religion and magic, as well as with natural prognostics. Iamblichus's treatise has a history of continuous reception in European thought, with revivals in the Renaissance, from Byzantium to Italy and France, and renewed in the 18th century in England and beyond. Scholarship on divination in Antiquity, and its philosophical conceptualisation, as well as its integration with religious thought, flourished in recent decades, as the new critical edition and translation of Iamblichus' key work, *De Mysteriis,* shows ([Clarke et al. 2003](#)).

Central to my focus is the establishment of Bantu divination (traditionally connected to African pre-Christian or pagan religion) in modern South African society. However, in some branches of Southern African Christianity, it is seen as 'prophecy', as Bengt Sundkler showed in a study on Bantu prophets in African Initiated Churches, and has

received new recognition here (Sundkler 1961, p. 277ff). Divination also continues to be practiced in pagan contexts. Interestingly, Bantu divination was also received in the field of psychotherapy, notably in the psychoanalytic school of C.G. Jung. Its integration of the psychological and spiritual raised particular resonance. This contributed much to the recognition and establishment of Bantu divination in the psychotherapeutic field. In this way, the spiritual and religious aspects of Bantu divination came to be received here too—in the context of somewhat secular modernity, nonetheless with spiritual interests.

The wider context is the spread of (elements of) African Traditional Religion in Modernity beyond Africa, in Latin America, but also in the Western 'North'. The spread and revival of African Traditional Religion, and its derivatives, in modern societies, is of particular interest. This phenomenon is observable in Africa and in Latin America. Some 'outliers' have also spread to Europe and North America in the north, via emigrant communities or through cultural contact. Several processes are involved. One is the decline of African Traditional Religion (ATR) through the advent of Christianity and Islam (Iliffe 1996, p. 37ff.)—counterbalanced to some extent by the development of syncretistic forms of Christianity, which integrate essential features of ATR. This happened through related processes of 'spread', through missionary work or in some cases through conquest, and in Latin America through social compulsion and through conversion, individually and collectively.

Mostly restricted to the early colonial era (Delgado 2019), it extends also to somewhat later times. The relations between colonialists and missionaries were often ambiguous, with collaboration or promotion on the one hand where a common ideal of Christian civilisation was assumed, and critique or conflict on the other, where Christian and colonial value systems and practices clashed (Hochschild 1998, p. 262ff.). Here, colonialists, African traditionalists, and Christian missionaries appear in different constellations of conflict or alliance, as vividly depicted by Hans Merensky in his report on his work as a missionary in this period (Merensky 1996, p. 458ff.).

The other process is the rise of secularism in the wake of Enlightenment. (It sometimes led to tensions between colonial authorities and missions, where secular naturalism and capitalist orientation clashed with religious values). In view of the secularism and naturalism of modernity, the African traditional world view is merely superstitious—to be overcome for the sake of progress. This third process of secularisation is presently working against all religion world-wide, with few exceptions (Stolz and Hackett 2023).

In view of these processes, the establishment of Bantu divination, in the fields of therapy (as a legally protected and regulated profession), psychotherapy, and by its rebirth in 'syncretistic' forms of African Christianity in the context of modernity and urban culture, is astounding. It is safe to declare that in this way a distinct form of Southern African modernity is emerging. This is supported by its study (and recognition as valid system of thought) in scholarly literature and in poetics.

It is comparable to Brazil, where Bantu divination emerged in Modernity, in the fold of Umbanda (Kleinhempel 2018a, p. 71ff.), notably in modern (and middle class) urban areas of Brazil well beyond its traditional ethnic basis (Kleinhempel 2018b). Complex processes were involved in this 'inter-cultural' or trans-cultural' reception and emergence. In brief: the core element of Bantu divination became enveloped in ritual forms and theology, strongly influenced by Nigerian Yoruba religion. Thus, 'Umbanda' was formed. It interacted with the spiritualist spirits of Kardecism that had emerged in France at the end of Romanticism. Its roots in divinatory traditions go back to Neoplatonism in Antiquity (Kleinhempel 2021). Allan Kardec wrote a systematic exposition of mediumism (Kardec 1861), which became a handbook on divination. Kardecism spread to Germany, to other European countries, and finally to Brazil. Its reception was enhanced by the spiritual spiritism of African traditional world view, present in Brazil in Afro-Brazilian religions. The emergence of Umbanda, to become a religion with a core of Bantu divination, and an esotericist religious philosophy, was sparked by its reception in the context of Kardecist sessions in Niteroi, Rio de Janeiro, in 1908 (Linares et al. 2017, p. 21f.), commemorated as a 'founding event'

(Saidenberg 1978) by its emergence in the (then predominantly) white educated milieu, where Umbanda has established itself since. The decline of Kardecism in Europe was partly due to the re-interpretation of its spiritistic phenomena, as 'manifestations of the unconscious' in a psycho-analytic sense (Saraceni 2017, p. 97ff.), thus revealing a link of reception by re-interpretation that is also observable for the encounters of some Jungian psychoanalysts, such as M. V. Bührmann, with Bantu divination. (It is an ambivalent relationship of 'appropriation, as will be shown in Section 5.2).

This excursus may shed a light on cultural dialectics between European Esotericism—of which C.G. Jung shared some concepts—and Bantu divination. The resonance it raises in European Esotericism, Jungian psychoanalysis, and older traditions of divination, of Graeco-Roman antiquity, will be shown to have facilitated the emergence of Bantu divination in the context of modernity in South Africa, and similarly in Brazil. The fascination raised by its sophisticated mediumism—also adopted in Umbanda in ritual forms (Vieira and Saraceni 2009, p. 64ff.) and theory (Saraceni 2017, p. 97ff.) —motivates its reception in these 'milieus' on both sides of the Atlantic. There is, however, a tendency to detach Bantu divination from its traditional African world view by 'European' reinterpretation, as will be discussed.

A view will also be taken on the corresponding reception of the African traditional world view supporting divination in South African Afrikaans, and in Brazilian literature of modernity, in the quest for a distinct poetics beyond the confines of European rationalistic naturalism. The perceptive abilities and the phenomenology accessed by Bantu divination are, in both cases, appreciated as a basis for a literature of Phantastic Realism, as poetic expression for a world view that integrates the 'spiritual', in post-secular forms of culturally distinct 'plural modernities'. (The latter concept contradicts the concept of European Enlightenment, which claims 'universal validity' based on its rationalism. It creates a tension also observable here).

The reception of Bantu divination beyond its black African ethnic base in South Africa by initiates from the white segment of the population (Kleinhempel 2018b, p. 148f.), as well as from abroad (Hall 1994), is a sure indication that here a distinct culture of 'modernity' is developing, with ramifications in the fields of the therapeutic, the legal system, the religious, and the spiritual. It is attractive in the 'Global North' too, especially in the West. (It indicates interactions between different forms and cultures of modernity). I investigated this spread, and the impact of the cultural reception of Bantu divination, in this realm in a previous study (Kleinhempel 2019a).

In this paper, I propose that specific interactions between strains of European thought—some of them marginalised or repressed by secular Modernity—and ATR facilitated the renaissance of divination and its establishment in modern societies in the Afro-realm and beyond.

## 2. Methodological Considerations

The perspective of investigation is shaped by the theory of syncretism, developed by Ulrich Berner (1982), by discourse analysis, systems theory, and by the history of religion and philosophy. The basic assumption is that cultures and religions form systems. These are autopoietic: they are not fully 'closed', as doctrinal debates in Christianity or in philosophical traditions show. Nevertheless, sets of beliefs, cultures, and values exist that allow one to speak of 'Bantu Philosophy', German Idealism, depth Psychology, or Roman Catholicism, etc. In a living process they interact with their environments and tend to adapt and reconstitute themselves, thus preserving their identity and fundamental views. According to Berner, 'syncretism' is not a random mixing of elements of religions, but a systemic process of reception and integration (Berner 1982, p. 83ff.). For conceptualisation, he draws on the systems theory of Nicklas Luhmann (1972). This means that the transition of Bantu divination into a modern society, which is predominantly Christian or secular with some influence of Esotericism, involves systemic processes of reception. Something similar can

be applied for the reception of Bantu divination into the realm of psychoanalysis, especially of Jungian depth psychology because of its metaphysical, partly esotericist, elements.

Modern South African society is 'Europeanised' to some extent. Therefore, the emergence of Bantu divination in the fields of Christianity and of psychotherapy can be regarded as a 'syncretistic' reception, rather than the (mere) 'survival of pre-modern tradition', even if it also has traits of such survival by adaptation. The difference lies with the reception: By taking interest in the metaphysical world view of African tradition, and the divinatory phenomena and practices embedded in it, this complex becomes appreciated—and reinterpreted—by the 'receiving formations'. This can be shown for branches of African Initiated Christianity, of the 'Zionist' type (Kleinhempel 2019b, p. 17ff.), where Bantu divination is adopted and endorsed as 'prophecy' (Sundkler 1961, p. 261ff.), with some adaptations. A similar process will be shown for South African psychoanalysis and therapy. The inclusion of this field here is warranted not only by the religious aspects of Bantu divination, but because Jungian depth psychoanalysts took interest in it, on the basis of the spiritual concepts in Jung's thought. Thus, a certain de-limitation between the aspects of 'religion' and of 'psychology' will be observed in this process of (systemic) reception, with features of 'syncretism' by the correlation of the two systems involved here. This involves a view to the hermeneutics applied in these inter-cultural encounters and understanding. Sensitivity to these perspectives with regard to their contexts has grown in recent years (Janz 2015, p. 478ff.).

### 3. The Re-Emergence and Establishment of Bantu Divination in Modern Southern Africa

The resurgence of Bantu divination in Southern Africa happened in two forms: the one is the integration of divination by 'African Independent Churches' of the 'Zionist' type (Sundkler 1961, p. 177ff.). This process is often conceptualised as one of 'syncretism'. The other is the emergence of divination in contexts of modern society, also among professionals and academics. This type is defined by tradition, in roles such as 'Sangoma' or 'iGqirha', as Nomfundo Mlisa describes in her anthropological dissertation on her initiation and its traditional form (Mlisa 2009).

The public recognition, regulation, and protection of the profession of Bantu divination is enshrined in law in South Africa, as proclaimed in the *THP (Traditional Health Practititioners) Act No. 35*, 2004 (The Presidency 2004).

Its purpose is as follows:

"To establish the Interim Traditional Health Practitioners Council of South Africa;

to provide for a regulatory framework to ensure the efficacy, safety and quality of traditional health care services;

to provide for the management and control over the registration, training and conduct of practitioners, students and specified categories in the traditional health practitioners profession;

and to provide for matters connected therewith." (ibid.).

This to ensure the quality of the professions comprised herein, and the supervision, the maintenance of standards, the public recognition, and the delimitation of fields of practice. It recognises traditional institutions that provide for the training and qualification of a range of traditional healing forms, among which are the diviners. The passage quoted here with its definitions is interesting for its explicit recognition of the role and status of the diviner as one of the professions of 'traditional healing', and of 'traditional philosophy', even mediumistic communications from 'ancestors', thus providing the concept of 'ancestors' with legal recognition, and the practices based thereupon as therapeutic interventions.

"Definitions:

1. In this Act, unless the context indicates otherwise -

'accredited institution' means an institution, approved by the Council, which certifies that a person or body has the required capacity to perform the functions

within the sphere of the National Quality Framework contemplated in the South African Qualifications Authority Act, 1995 (Act No. 58 of 1995) (...)

'Council' means the Interim Traditional Health Practitioners Council of South Africa established by Section 4; (...)

'diviner' means a person who engages in traditional health practice and is registered as diviner under this Act; (...)

'health services' includes inpatient or outpatient treatment, diagnostic or

therapeutic interventions, (...)

'herbalist' means a person who engages in traditional health practice and is registered a herbalist under this Act; (...)

'traditional health practice' means the performance of a function, activity,

process or service based on a traditional philosophy

that includes the utilisation of traditional medicine or traditional practice and which has as its object -

(a) the maintenance or restoration of physical or mental health or function; or

(b) the diagnosis, treatment or prevention of a physical or mental illness; or

(c) the rehabilitation of a person to enable that person to resume normal

(d) the physical or mental preparation of an individual for puberty, adulthood, (...)

'traditional health practitioner' means a person registered under this Act in one or more of the categories of traditional health practitioners; (...)

'traditional philosophy' means indigenous African techniques, principles, theories, ideologies, beliefs, opinions and customs and uses of traditional medicines communicated from ancestors to descendants or from generations to generations, with or without written documentation, whether supported by science or not, and which are generally used in traditional health practice; (...)." (The Presidency 2004, p. 6).

It is interesting that both physical and psychological sickness and health are mentioned, along with diagnosis, prevention, treatment and rehabilitation; also mentioned are—by allusion—the rituals of life transition as contributing to well-being. African traditional philosophy and beliefs are recognised explicitly as sources for its practices, including mediumism and healing rituals They are endorsed as a legitimate basis of divination and of other forms of traditional African health practice. This includes the religious aspects of divination. This means that those who practice them are legally entitled to do so and are protected with accreditation by the Interim Traditional Health Practitioners Council required. (The latter is important for the quality and status of the profession, and as a safeguard against quacks, who bring these professions into disrepute). By this law, the religious beliefs, rites, and practices relating to physical and psychological health and well-being are included and recognised as legitimate therapeutical forms. It includes mediumism and divination.

The tasks of the governing body are defined as follows:

"Establishment of Interim Traditional Health Practitioners Council

4. (1) A juristic person to be known as the Interim Traditional Health Practitioners Council of South Africa is hereby established. (...)

5. The objects of the Council are to (...)

(d) promote and maintain appropriate ethical and professional standards required

from traditional health practitioners;

(e) promote and develop interest in traditional health practice by encouraging

research, education and training;

(f) promote contact between the various fields of training within traditional health practice in the Republic and to set standards for such training;

(g) compile and maintain a professional code of conduct for traditional health practice; and

(h) ensure that traditional health practice complies with universally accepted health care norms and values." (The Presidency 2004, p. 8)

This indicates that the government is working towards a professionalisation of traditional healing, including divination, to ensure standards of quality and to promote further development.

The extent to which traditional divination is enrooted in society may become apparent from an investigation amongst township youths, as to which proportion of them have consulted sangomas or intend to do so. The authors declare the following:

> "Since the year 2000, the presence of traditional healers has been progressively noted. (. . .) Older people tend to approach them for a range of different reasons (. . .) However, the extent to which younger people draw on the services of sangomas has been a point of debate." (Nyundu and Naidoo 2017, p. 146).

In their representative study on the attitudes of youth in townships, Tony Nyundu and Tony and Kammila Naidoo found that approximately 30% of the youth respondents declared to have consulted a sangoma already, and more than a third would do so in future (ibid., p. 151). Forty percent would respond positively if they received the 'call' to become a sangoma themselves (ibid., p. 152).

On the homepage of the South African Healers Association (SAOHA), Prof. Velaphi Mkhize declares:

> "South African Healers Association (SOAHA) is a community of healers, spiritually, physically and intellectually, engaged in the task of influencing the development and support of various healing practices in South Africa and recognizes the spiritual elements of these spiritual, traditional, indigenous and natural healing practices. One common theme in research activities is that all relate to the accumulation of knowledge, whether scientific or unscientific, theoretical or practical.
>
> SOAHA is a fully registered NON-PROFIT COMPANY (NPC), with the office of Company Registry in Pretoria, with the aim of registering all interested Multifaceted Healers Provincially, Nationally and Internationally for the sake of promoting African Healing through Research, Education and Publications.
>
> SOAHA has identified Five Pillars within which it will operate:
>
> Research and Education (Create cohesion and Relationship between Indigenous Healing and Modern Medicinal Healing Practices and Models.
>
> Spiritual Healing and Transformation
>
> Ancient Wisdom and Interconnectivity
>
> Connecting the Past to the Present to forecast the future
>
> Connection of Indigenous Healing Models to the Modern Healing Practices." (Mkhize 2013).

This gives an indication of the process of organising the traditional healers and diviners of South Africa, with the process of formation and professionalisation in the contexts of the legal system, the academic sphere, and the therapeutic and spiritual realms. The integration of the spiritual, also the mediumistic, and the religious with the therapeutic, and the stated intention to connect them with other forms and traditions in the modern world are important.

### 4. Divination in European Tradition and Its Relevance for the Reception of Bantu Divination

Divination is deeply enrooted in European cultures. This is important to our investigation for systematic reasons.

Firstly, Modern African societies are strongly shaped by the European cultures, historically during colonialism, but also continuing at present through the use of European languages, the adoption of academic traditions, value systems, discourse formations, etc. To a certain extent, modern sub-Saharan societies form part of a wider cultural 'Euro-sphere'. This means that European philosophical traditions in particular form part of the heritage of modern African cultures and societies. It is comparable, in a way, to the reception of the cultural and philosophical legacy of Greco-Roman antiquity by the hitherto illiterate peoples of central, northern, and eastern Europe. From this synthesis, modern European cultures developed. The process occurred mostly after the period of Roman colonialism, in some of these regions, and continues to this day, to some extent. The same may be stated for modern African cultures.

The most important 'philosophical theoretician of 'divination' in Greco-Roman Antiquity, Iamblichus of Chalcis, discussed the religious aspect of divination and current naturalistic attributions as follows:

> "... to know the principle of divination, to know that it is activated neither by bodies nor by bodily conditions, neither by a natural object nor by natural powers, neither by human disposition (...) Divination ... consists of divine vision and scientific insights. All else is subordinate, instrumental to the gift of foreknowledge sent down by the gods: everything that concerns our soul, our body, everything inherent in nature of the universe ... (...) if someone ... downgrades the skill of divination to secondary operations—position for example, bodily movements or changes of emotions ... either activities of human life or other psychic or physical explanations—he might believe that he says something obvious. (...) There is one ... first principle concerning all these matters: that is, ... to derive it from the gods who in themselves possess the limits of all knowledge of existing things, from whom the mantic power is distributed throughout the whole cosmos, and among all the different natures found here."

(Iamblichus, *De Mysteriis* III. 1.)

In this way, he firmly declares divination to be of a 'transcendent' quality and origin that surpasses the regular spatio-temporal order of the world. The reference to the 'gods' may be read from different perspectives. Assuming one supreme divine origin of the whole cosmos, Iamblichus conceives the 'gods' as 'higher intelligences' that embody specific aspects of the divine and possess transcendent knowledge that they impart to humans and to other living beings as well. This implies that divination is not conceived of as a function of ordinary nature, but also not a specific religion, but as a universal divine gift or endowment, present and active in the universe. He acknowledges the influence of somatic and psychological factors, but rejects any naturalistic attribution of divination. This definition remains with the concept of divination, relating it to a religious world view, but also distinguishing it from religious ritual. This feature renders divination accessible to modern theories that seek to locate 'the transcendent' in a wider concept of the 'soul', particularly by C.G. Jung, with his concept of the 'collective unconscious', which facilitated the reception of Bantu divination in modernity, especially in South Africa. The role of physical, bodily, psychological, and emotional factors is acknowledged by Iamblichus, but subordinated to the transcendent. Thus, he continues by discussing divination through dreams (Iamblichus, *De Mysteriis* III. 2.).

Divination was a respected cultural institution throughout Greek and Roman Antiquity, with famous oracular sites such as at Delphi in Greece, and with philosophical literature reflecting on divination, conceptualising it metaphysically and defending it culturally (Plutarch: *De Defectu Oraculuorum*, Plutarch 1936). The Middle Platonist philosopher

and historian Plutarch of Chaeronea described the institution of divination, as practised by the oracle of Delphi, and its relation to society and culture. Several eminent sites of divination were visited over centuries for oracles on personal, communal, and theological questions (Copenhaver 1992, p. xxiv). This institution is even discussed briefly in the *New Testament (Acts* 16:16ff.). The extensive description of a prominent case of divination in the *Old Testament*–King Saul's visit to the seeress of Endor (*1. Sam* 28: 3ff.) attests to its recognition by the priestly authors in the time of the Babylonian exile of Israel, presenting it as a most ancient institution and affirming its effectiveness (Eissfeldt 1956, p. 367). It is thus embedded and recognised in the Christian consciousness. It is also a firm part of the traditional cultures of northern Europe (Ryan 1999, p. 70f.), described, e.g., by Tacitus for the Germanic tribes (Tacitus: *De origine et situ Germanorum,* Tacitus 1942).

From the Middle Ages, through the Renaissance, and up to Modern times, an awareness and appreciation of divination is documented, even in 'high-cultural' literature. François Rabelais's treatment of the issue is an eminent example. As a canonical text of French literature, it has authority. Rabelais sums up authors of Antiquity on the issue of divination here through dreams, in the third book of his five books on the deeds and sayings of Gargantua and Pantagruel:

> "…let us bend our course another way, and try a new sort of divination. Of what kind? asked Panurge. Of a good ancient and authentic fashion, answered Pantagruel; it is by dreams. For in dreaming, such circumstances and conditions being thereto adhibited, as are clearly enough described by Hippocrates, . . . by Plato, Plotin, Iamblicus, Sinesius, Aristotle, Xenophon, Galen, Plutarch, Artemidorus, (…), and others, the soul doth oftentimes foresee what is to come (…) such a one as by the Greeks is called onirocrit, or oniropolist. (…) The sacred Scriptures testify no less…." (Rabelais 1545, III. 13)

Rabelais renewed the knowledge of divination with his extensive list of authors of antiquity who wrote about oracles and divination. From here, a line of tradition may be pursued to the 19th and early-20th centuries, with their revival of interest and studies of this tradition, of which the co-founder of analytical psychology, Carl Gustav Jung, is part. From the mid-20th century on, the field also received intensive scholarly attention. Jung gained essential insights from it, and wrote favourably about the faculties of pre-cognition (Jung 1951, p. 280f.). He drew on some of the same authors as Rabelais. Thus, he provided a foundation for psychoanalysts and psychiatrists of his orientation to take interest in Bantu divination and to appreciate it. A 'dialogue of traditions' is emerging here that needs to be continued.

In view of this heritage, it is no surprise that Europeans should resonate with the institution of divination in Bantu cultures, as practiced through sangomas (or igqirhas, the Xhosa term for diviners). The resonance is different from the ethnographic studies of divination in non-European cultures, which treat the issue as a phenomenon of the 'others', under the perspective of alterity. They may be respectful of it, even sympathetic, as in Mircea Eliade's influential book on shamanism, which brought the complex cosmology, and functions and powers of the shamans in different societies, to the awareness of a wider academic readership in the twentieth century (Eliade 1975, p. 177ff.). Although Eliade focussed on north and central Asian shamanism, drawing on a rich Russian ethnography, he also included perspectives on European, American, and southern Asian traditions too. However, a perspective on Africa is missing. (It may reflect the view that African divination has little to do with Siberian shamanism. In view of phenomenological similarity, the distinction between African and 'out of African' cultures may be questioned from a systematic perspective).

## 5. South African Psychoanalysts as Protagonists for the Reception of Bantu Divination

### 5.1. Bernard J. F. Laubscher (Psychiatrist)

In South Africa, psychoanalysts encountered Bantu mediumism, both in their patients and by diviners. They wrote about it from the early-20th century onwards, with Sigmund

Freud's interest in 'pre-rational' thought patterns in mind. Thus, they contributed to the emerging field of 'ethno-psychiatry' (Pringle 2019, p. 8). As may be imagined, the issue of cultural differences was burdened with political implications, depending on how it was framed. Differences in world view could easily be charged with connotations of 'pre-rational', thus devaluating them. Two approaches emerged, either to reject any world view differing from that of European Naturalism as delusional, sometimes condescendingly called 'primitive', or to accept it. (The issue is reflected by pioneering authors in the field, such as Placide Tempels, in his *Bantu Philosophy* (Tempels 1959). The realm of psychiatry is highly interesting in this regard, because here, perceptions and experiences beyond the pale of 'Western' rationality were encountered which had to be diagnosed, either as delusional or as culture-specific, or as a combination of both. It raised diagnostic challenges that are still contested.

A foremost 'ethno-psychiatrist' in the early-to-mid-twentieth century was the South African Bernard J. F. Laubscher (1897–1984). He studied in Glasgow, where he became familiar with psychoanalysis and psychic research. Later on he went to the United States as a research fellow. Working in a rural area of the Eastern Cape province, he encountered the influence of the African traditional world view on psychic conditions among his patients. In *Sex, Custom, and Psychopathology: a Study of South African Natives* (Laubscher 1937), he presented his findings in a pioneering study on cultural and religious factors there. Through his own observations as a participant in mediumistic sessions in Scotland he had, and empirical familiarity, he was prepared to accommodate mediumistic phenomena, experiences, and perceptions, conceptualised in the Xhosa world view. He had the professional expertise and an open mind-set to distinguish psychiatric aspects from those of Bantu spiritism, and to reflect on their interrelation in syndromes, which he encountered amongst individual patient cases (ibid., p. 220f.). Laubscher was knowledgeable, and not only in psychoanalytic theory, which he applied astutely in diagnostics and treatments, located in a psychiatric hospital in the rural town of Queenstown, South Africa. He was also familiar with research on the limits of consciousness and the nature of the mind, which flourished in the first decades of the 20th century. In a review of it, Edward F. Kelly and co-authors declared:

> "[William] James pointed out that to describe the mind as a function of the brain does not fully specify the character of the functional dependence. (...) More generally, one can at least dimly imagine some sort of mental reality, in which James's view might be anything from a finite mind or personality to a World Soul, that is closely related to the brain functionally, but somehow distinct from it. (...) Like James (1898/1900) and McDougall (1911/19861) among many others, I will immediately appropriate the entire body of evidence for psi phenomena in service of our central thesis." (Kelly et al. 2007, p. 28f.).

Laubscher alludes variously to this model of a 'mind' or 'consciousness' that is more than a function of individual brains to explain the mediumistic phenomena that he encountered. He also refers to the preparation he had received in the occult subculture of Europe, and in the United States:

> "Among them were university graduates all indulging in the dark emotional strata of the unconscious. Did the pagan in his red blanket have a collective unconscious memory of a psychic experience once lived in conflict with those darker forces of the mind? (...) But then there is that fascinating unseen world from which an array of mythical characters emerge and again disappear into the 'nothingness' where physical eyes cannot follow. But once the pagan had found that one understood the psychic life, then the barriers fall away." (Laubscher 1975, p. 12)

In this passage, Laubscher refers to established concepts and to factors influencing his communication and observations in the field: First of all, he reminds the reader that paranormal practices and phenomena were also experienced by European and American

academics. Thus, he refers to the spiritist 'subculture' which flourished from the 18th century onwards, up to the time of his writing, the 20th century. He reminds the reader that this is part of European and American culture, albeit as a subculture, and not limited to the 'un-enlightened' parts of the world, at the 'fringes of civilization' (Pons 2017, p. 77f.). His argument is based on its establishment in the upper, academic tiers of societies; thus, one of cultural prestige. In this context, it appears as necessary to fend off the racial-cultural bias between allegedly 'rational' Western culture and the 'superstition of the primitives'. (The "pagan in the red blanket" refers to the ochre and red mantles worn traditionally in Xhosa culture). It is a figure in the background that emerges repeatedly, also with other authors. It reveals tension in his thought, on which he reflected. But he overcame his reservations through close observation and encounters in his psychiatric work and field work. He analysed the interplay between psychological disorders and the paranormal in detail, yet remained careful to distinguish them categorically. He took cultural values and social order into account as well, thus making his work of lasting importance. On this basis, he distinguished between divination and magic, and also looked at the shady sides of the spectrum:

> "I have been especially impressed by the honesty of the isanuses [senior diviners] in their description of the extent of their psychic abilities. They do not claim any powers of magic and modestly admit that they can only see and tell in so far as their minds are opened to the influence of the onomathotholo [guardian spirits]. [...) The amaxhewele are the quacks of the profession, and they frequently claim and diagnose ukutwasa conditions [the mediumistic trance that manifests a call to divinership, and is not voluntarily attainable]. They also have the various evil medicines, medicines by means of which they can bewitch people.... [...] They obtain this power by consorting and working with evil influences of witchcraft and magic. They are despised by councillors, elders and by the isanuses. The line between the amagqira [diviners] and the amaxhwele is not quite distinct..." (Laubscher 1937, p. 35).

The above shows which challenges are connected with the framing of traditional Bantu divination in South African law, as a healing profession. It underlines the need for supervision by traditional bodies, to safeguard the ethical and professional standards in this field.

About the condition of 'ukutwasa' (the trancelike state that indicates a mediumistic calling, and inaugurates it), which has psychiatric aspects and is sometimes confused with psychotic disorders, he writes:

> "The isanuses, igqira and ixhwele [the herbalists who also practice magic] all maintain that uktwasa can be hereditary and, if not properly treated at its first manifestation, may lead to a loss of senses. The amagqira and amaxhwele point out that ukutwasa stares may appear in many family histories and over many generations. [...) the isanuses and amagqira equally assert that the capacity for the development of psychic powers, manifested in its incipiency as ukutwasa, is inborn and cannot be developed by any form of training if the gift is not possessed by a person..." (Laubscher 1937).

Laubscher frames these phenomena not as 'religious' but as 'psychic' and distinguished them from the 'psychological', especially in view of symptomatic overlap.

About the reality of paranormal faculties with diviners, Laubscher reports extensively of his own tests, to which he put a befriended isanusi, Solomon Daba, about objects that he had hidden. Summing up the result, Laubscher stated:

> "I requested as séance dance and told him that I had prepared a test. During the dance, Solomon Daba described in minute detail the article, the nature of the locality in which it was buried the brown paper in which it was wrapped ...[etc.] ... I never once gave any information ... It will be assumed that he was reading my mind telepathically. ... he accomplished a remarkable feat and

displayed supernormal mental abilities. This is only one of the experiments . . ." (Ibid., p. 43).

Laubscher conceptualises this further, with regard to its therapeutic application:

> "I have no doubt that these isanuses and amagqira have a form of psychic power which one may call psychic sensitivity, because the word 'intuition' does not cover all its phases. (. . .) certain tests carried out on amagqira show that they can give a fairly accurate picture of what a person has in mind, especially those ideas having strong emotional values. This psychic function is commonly employed in making their diagnoses." (Ibid., p. 45).

As to spiritual aspects of divination, he alludes to the self-concept of European Spiritism, that flourished, especially from the mid-19th century on, in Kardecism. Matthias Pöhlmann explains its focus and delimitation:

> "The central themes that Spiritism took up and articulated in its practice were death and the otherworld, the postmortal fate of the dead in the spirit world, but also relations between the living and the dead. In so doing Spiritism, in a secular cloak, picked up forgotten or suppressed themes of Christian theology and ecclesiastical proclamation." (Pöhlmann 2004, p. 54)

Laubscher encountered these 'beings from beyond' as 'agents' in the lives of his patients, and as entities with whom the traditional diviners dealt with. With critical reserve, he nevertheless acknowledges the phenomenal presence and 'agency' of spirit entities.

About the religious significance of the spirits entities—especially the 'ancestors' that are invoked as guardian spirits—he writes:

> "The pagan native . . . is frequently accused of worshipping his ancestors, but this is not the case . . .) Above them is another higher order, which existed long before their births and . . . is synonymous with our idea of God. (. . .) He maintains that this higher existence gives life and makes things grow. Hence he sacrifices and appeals to his ancestors, who are nearer this order [to intercede for him (. . .) Now all that is good comes from this higher order and works through their ancestors . . ." (Laubscher 1937, p. 55)

As to the spirit entities, especially with the ancestors, who are invoked as guides and mentor spirits, Laubscher relates to a concept of C.G. Jung: he frames them as emanations from a 'collective unconscious'. Jung defined this concept in a treatise on the concept of the Collective Unconscious, (Jung 1984a, p. 114) on which he had lectured in London at about the time of Laubscher's field studies for this book. Jung declared that iconic figures occurring trans-personally, and yet in individual dreams of imaginations, as well as in collective 'imaginary realms', are emanations from an inherited collective stratum of the psyche. He implied that there may be different layers, from common inheritance of humankind up to culturally specific forms. By adopting this concept, Laubscher reframed the phenomena and perceptions made individually by his patients, and yet held collectively by the Xhosa people. He legitimised the perceptions of spirit entities in the frame of (modern) depth psychology. On this basis, a degree of reality—beyond mere individual and culture-bound delusions—was ascertained for them.

Laubscher followed a figure of thought here that seeks to reconcile the idea that spirit beings are a projection of the (collective) psyche, and the acceptance that they may have some ontological reality and agency of their own. (Jung shared this to some extent). Wouter Hanegraaff explains that in 'New Age' thought, a dualistic premise is rejected, which

> "distinguishes an objective metaphysical reality from merely subjective impressions belonging to the psyche (. . .) that the distinction between objective reality and subjective experience is absolute. . .. Without this premise the relation between the psyche and od, but also between the psyche and e world, is seen in a different light. It is no longer possible to take . . . distinctions between reality and 'mere imagination' . . . for granted." (Hanegraaff 1998, p. 225)

His view is shared by Jung, who included the religious in his view of the individual and collective psyche (ibid., p. 508). Laubscher did not limit himself to studying this realm of Bantu world view and its influence on his psychiatric patients. He befriended a Xhosa diviner, Solomon Daba, studying concepts and practices of divination through him. The figure of the 'collective unconscious' helped Laubscher to conceptualise both his observations and the inner resonance, which he felt during sessions. His report on the experiences with Daba reveals both professional interest in the process of diagnosing a patient, and personal sharing on a common basis as therapists (relevant for the reception of Bantu divination as a 'healing practice' subsequently, especially in the therapeutic field). Laubscher writes:

> "Then one day Solomon Daba very casually mentioned an association between ukutwasa, the development of mediumship, at the call of the Abantubomlambo [the 'River people' spirit beings, whom Laubscher regarded as having both aspects of the super-ego and of descent into the deeper layers of the psyche], and my mind at once seized this as a new and valuable lead. I was informed that during ukutwasa some people dream, about speaking to their ancestral spirits. (...) It is at this stage [in the training to become a diviner, an Igqira] when things are revealed to one inside one's head. One talks to a person who is ill, one just knows where he came from and how he feels, where pain or discomfort is. (...) Solomon Daba described all this in a most serious and logical manner. He was taking me on a mental journey into the hidden world of Xhosa thought. The feelings and perceptions which underly the awareness of the pagan mind and give it a wisdom which raises him and gives him the distance of dignity and makes him feel the satisfaction of a deeper contact with life which somehow means more than the knowledge and power of other people. I for one could not help becoming aware of some common universal level of consciousness in which our thinking was having its existence. Indeed that we were in tune on a certain universal level of the Cosmic Mind. The unison of understanding transcended his red blankets and my European clothes, my education and his illiteracy. We were in contact with a stratum of the cosmic consciousness. (...) The Xhosa Isanuses [senior diviners] describe many facets of this strange experience of ukutwasa. The chief characteristic however remains an awareness of things and events far beyond the world of our senses. It was at that moment that an intuitive flash like sheet lightning in a Transkeian night suddenly lay bare a landscape of psychic significance." (Laubscher 1975, p. 25)

It is well worth looking at the motifs, the concepts, the experience, the perspectives of writing, and the discursive means in this quote. Laubscher merges three horizons: his own perspective, as a psychiatrist and psychoanalyst of Jungian orientation; the world view of Xhosa culture and diviners, with its spirit beings (whose ontological quality he accepts, despite his psychoanalytic interpretation of their symbolism); and the esotericist notion of a 'Cosmic Mind', which is indebted to the Platonic philosophical tradition (Hanegraaff 1998, p. 120f.),—Laubscher seems to draw on its concept of the 'World Soul', with the understanding that the individual souls participate in it (Karamanolis 2020). Furthermore, he mentions the Theosophic concept of 'Akasha record' (Laubscher 1975, p. 36), as a store of information about things past or to come (Brandt and Hammer 2013, p. 122f.). He also refers to 'Hindu' metaphysics (Laubscher 1975, p. 196) to explain some phenomena of 'subtle energy', an 'astral world'. In this triangle, Laubscher situates S. Daba and himself in a 'merger of horizons' (George 2021), striving for mutual elucidation of his observations, and experience in this encounter.

It is interesting that Laubscher included his own thoughts and feelings in this encounter. It may be due to his training in psychotherapy, to pay attention to himself and his inner processes in such encounters as relevant to understanding. This agrees with the emphasis on hermeneutics as proposed by Wilhelm Dilthey for 'understanding'. It includes, by necessity, the subject, as the 'participant observer': "Dilthey associates the purpose of

the human sciences not with the explanation of 'outer' experience, but, instead, with the understanding of 'lived experience' (*Erlebnis*)." (ibid.). This is eminently important for the reception of Laubscher's description. Vicariously, he enables the educated reader to identify with him, and to approach this realm from his perspective (feeling with him and seeing with his eyes).

The pathway of Jungian psychoanalysis has been adopted, subsequently, by other authors in the field. Thus, he provides access for the white South African reader, but also for the black reader seeking to approach this heritage with a professional academic view. This was continued by the (white) Jungian psychoanalyst Dr. Vera Bührmann (1986), the latter by the (black) academic psychologist Dr. Lily Rose Nomfundo Mlisa (2009) in their respective writings about the realm of Bantu divination, and also by others outside of the realm of psychology.

Laubscher's philosophical access of feeling connected to the realm of Bantu divination by a "universal level of the Cosmic Mind" complements his psychoanalytic view and relates to it. In a cultural and social environment of segregation, and power imbalance, in South Africa, Laubscher thus combines the cultural difference of world views and the experience of understanding and sharing. All this from within a philosophical and psychoanalytic frame that strives to make it acceptable to the readership, in particular the white reader. (This is also relevant to the non-South African 'Western' reader, who may feel aloof of the 'primitive' or 'non-rational' mind'). By including his own feelings and inner experience in his description of the encounter, Laubscher here frames the acceptance of Bantu divination in his own professional 'persona', with which the reader is invited to identify.

In his book *The Pagan Soul*, he gives a detailed report on the world view and divination of the Xhosa people (Laubscher 1975, p. 36ff.). He describes divination in detail, linking it with research on extrasensory perception and paranormal faculties in Europe and America, with the philosophy of Esotericism, and with ideas and practices of spiritual spiritism (ibid., p. 202f.) which flourished in Europe and the Americas from the mid-19th century on. Exploring its resonance and agreements with the Bantu world view (and divination), he inscribes the latter into these discourses. In this way, Laubscher presents the Bantu world view as confirmation of European spiritual spiritism, and its practices and concepts as meaningful elaborations, which surpass them in many instances and complement them. Laubscher presents the Bantu world view and divination as a means to retrieve what has been lost by Western rationalism, thus forming it into a critique of the limitations of modern secularism, and of (mainline) Christian theology:

> "Soon these pagan psychics will be gone. Western civilization, technology, school and the church and Homeland government must change all of the old order. (. . .) The . . . church could hardly be expected to understand Ukutwasa as the development of inner psychic faculties, or as the revelation of the consciousness of the innerself when the modern world . . . is oblivious to the dynamics of spiritual life.

> It is hence with an element of sadness that one sees the institution of the pagan psychic faculties falling into disuse, and being forgotten in the new things of a materialistic culture." (Laubscher 1975, p. 58)

In the meantime, however, Bantu divination has emerged powerfully in environments of Westernized modernity in Southern Africa, practiced by urban professionals as diviners and as clients, with widespread social acceptance. Similarly, Bantu divination has been adopted, with some adjustments, into African Initiated Churches (of Pentecostal type), which integrate much of it in their theology and practice as a form of 'prophecy', and as a spiritual charisma (Sundkler 1961, p. 260ff.).

### 5.2. Maatje Vera Bührmann (Jungian Psychoanalyst)

Following Laubscher, another South African psychoanalyst emerged who wrote about her encounters with Xhosa (Bantu) diviners: Maatje Vera Bührmann, and who expressed

her respect for them. She belonged to the Afrikaans people, coming from a noted family that had fought against the British. In this way, she belonged to the ruling caste in the times of Apartheid, when a brutal system of disenfranchisement and discrimination of all coloured people, including dispossession from land, ensured that political power came into the hands, mainly, of the Afrikaans segment of the white population. Her writings about Bantu divination thus had authority. André Landman, in his research on Vera Bührmann, focussed on her social position. He emphasises "the fact that she remained part of the Afrikaner establishment, a person with many influential contacts in Afrikaner intellectual and political circles." (Landman 2019). In addition to her background, she was a pioneer in the country by having a full training as a Jungian analyst in Great Britain (ibid., p. 100ff.) and by being "one of the prime movers behind the establishment of what is now the Southern African Association of Jungian Analysts (SAAJAs) (ibid., p. 1). Landman notes that her engagement for the establishment of Jungian psychoanalysis in South Africa happened at the same time and may have been connected in some way (ibid., p. 109). (It is a suggestion that I may support here). Unfortunately, Landman pays far more attention to her social standing and her position in the political landscape of South Africa in the mid-20th century than to her role as a pioneer of Jungian psychoanalysis and of studies of Bantu shamanism in the last decades of her life. His judgement, that both should be understood as supporting the racial discrimination politics of Apartheid in the last decades of white rule (ibid., p. 115) fails to discern that she had a keen sense of cultural collective identity and specific cultures, with which she appreciated South Africa's cultural diversity and embraced it. She reflected on this issue in a programmatic statement:

> "To quote Senghor yet again. 'We must in the 20th century enrich our civilizations through the mutual gifts and not create a new civilization'. I go with him only part of the way. Gradually a new civilization is bound to develop if the world is not set on a path to destruction. Those contributing to this civilisation should, however, remain in touch with their ancient and timeless roots." (Senghor 1977, p. 3)

This passage is interesting in this context, because Bührmann's reference to Léopold Sédar Senghor's philosophy of Négritude, with its dialectics of distinctness and interrelation with other civilisations, shows that she took interest in an Africanist perspective vis-à-vis the European. She shows herself to be critically aware of the dialectic forces of a sense of collective identity and of sharing in a multi-cultural society such as South Africa's, which she relates to Senghor's reflections on global processes. Landman criticised Bührmann for having an 'essentialist' view of culture, referring to Ken Roper's critique of Bührmann's work (Roper 1992). Roper criticised Bührmann for the literature that she drew on mostly, such as Berglund's *Zulu Thought Patterns and Symbolism* (Berglund 1976) and Sundkler's *Bantu Prophets in Southern Africa* (Sundkler 1961). This is strange, to say the least, as both are recognised authorities, with sound knowledge of Zulu culture. (Bengt. G. M. Sundkler worked as a missionary in Zululand for some years before becoming professor in Uppsala. Axel Ivar Berglund grew up as son of missionaries in rural Zululand, worked there, and had intimate knowledge of the Zulu language and customs. His work is referenced by authorities, such as David Chidester, on diviners (Chidester 2016).

He also criticised her for making generalisations from her fieldwork in one region of the Xhosa realm to the Xhosa in general and to Bantu culture more widely. However, he fails to take notice of the classic literature on Bantu philosophy, such as Placide Tempels (1959), and of John Mbiti (1969), also failing to recognise that Sundkler's work belongs to this category. Roper criticises Jung's interest in 'culture-specific' and 'trans-cultural' psychology as 'racist', and Landman follows him, also with regard to the work of Bührmann (Landman 2019, p. 114). What both Roper and Landman fail to reflect on, however, is that without the notion of distinct 'cultures' (and ethnic groups sustaining them), the concept of 'intercultural' relations becomes meaningless. The importance of distinct cultures, including religions, and often based on majority religions, as global factors in the sustenance and delimitation of collective identities, also in the political field, has been reaffirmed and

developed with wide resonance by Samuel Huntington in his *The Clash of Civilizations* (Huntington 1996) The labels of 'racism' (Landman 2019, p. 1), liberally applied to studies of specific cultures, as of Jung (ibid., p. 145), and of cultural 'difference', thus of distinction between the (collective) 'Self' and the 'Other', and also the label of 'essentialism' (ibid., p. 144) make such studies essentially meaningless.

Trans-cultural psychology has also come under criticism for not being focussed on the political aspects of inter-group relations in South Africa (Swartz and Foster 1984). However, a closer look, such as at the overview by Leslie Swartz (1986), often reveals a deep disregard for non-Western world views, and attempts to deconstruct ethnic group identities of distinct language groups and to eliminate the 'alterity' of the African traditional world view. The widespread motif, that reference to a collective ethnic identity and culture is 'racist', or 'colonial', may be rejected with a view to the description that the hero of South Africa's liberation struggle, Nelson Rolihlahla Mandela, gives of his own cultural roots, and how they shaped his destiny as a freedom fighter and a president. In his autobiography, he looks back to the origins of his 'tribe', the Thembu, and to the role of his family in its governance (Mandela 1994, p. 4ff). While mentioning the "white liberal sensibility" appreciatively, he declares that his own strength and authority emerged from his deep sense of his own culture and those of his people. Reflecting on his readings about a hero of liberation struggles in Europe, he writes:

> "One book that I returned to many times was Tolstoy's great work *War and Peace*.
> . . . I was particularly taken with the portrait of general Kutuzov. (. . .) Kutuzov
> defeated Napoleon precisely because he . . . made his decisions on a visceral
> understanding of his men and his people. It reminded me once again that truly
> to lead one's people one must also truly know them." (ibid., p. 478f.)

This statement mirrors Mandela's convictions about the value of his own distinct culture, also in the sphere of political leadership. In this spirit, the reappropriation of Bantu divination in the modern black African milieus of South Africa also has the aspect of affirmation of the own distinct cultural identity. The view of Nelson Mandela is interesting in this regard, as it indicates the inclusion of European discourses of cultural identity—here by Nikolay Tolstoy—for shaping attitudes to his own (non-Western) cultural heritage.

(It is a standard trope in South African inter-group discourses, that Anglophones who consider themselves 'liberal' or 'progressive', also in the academic sphere, tend to show disinterest in an African traditional world view and disregard for it, reinforced by monolingualism in a country of linguistic and cultural diversity. On this background, it is significant that the pioneering and best studies of African traditional religion were written by missionaries, by black Africans themselves—and here, in the psychiatric field, by Afrikaans authors).

This background indicates the eminent importance of the work of Laubscher and Bührmann, and of others in their footsteps, to overcome the disregard and disrespect of African traditional world view and the culture of divination. The recognition and close study of distinct phenomenology, perception, and experience in this realm, that Laubscher studied and researched from the 1930s on, remains an achievement that paved the way for the emergence of Bantu divination—by academic professionals—in modern South African society.

Viewed in the perspective of 'discourse analysis', the emphasis of cultural difference allowed for different political implications, including support for the politics of 'separate development' (Landman 2021). (It would be fallacious to denounce the studies of distinct African cultures because of the abuse of the notion of cultural difference for the legitimisation of Apartheid's brutal caste system of ethnic discrimination, which was euphemistically labelled as 'separate development').

It would therefore, be simplistic to reduce Bührmann's explorations of Xhosa divination to such political ambiguities. She had sought her own path to a wider world view, beyond the constraints of her own cultural background, and of Freudian thought, by becoming a Jungian psychoanalyst. Although she did not share the spiritualistic world-

view with Laubscher, she also sought to integrate religion and science of religion into her understanding of the cultures of her land. She did this, however, perceptibly on a more psychoanalytic basis:

> "The most important opportunity . . . which all racial groups in this country have is the achievement of a better understanding of one another by means of the psychological concepts of Jung and other authors, viz. Mircea Eliade (1960), on the history of culture and religion, Joseph Campbell on symbolism and mythology (Campbell 2012), and Victor Turner (1981) and Axel-Ivar Berglund (1976) on anthropology." (Bührmann 1986, p. 22)

Here, she proclaims the necessity to achieve a deeper understanding of all population groups in South Africa, by including history, religion, culture and anthropology, and to integrate these perspectives with those of analytical psychology. (This resonates with Jung's convictions). This integrative approach is significant, and enriching. The books that she annotated are classics. Some were published at the time of her field research. In particular, Victor Turner's *Drums of Affliction* reports on the transformations that he and his wife experienced as anthropologists during their field work in Zambia, studying phenomena of Bantu divination and their cultural and social contexts. It became clear to the Turners that their Eurocentric approach—specifically of Functionalism—was insufficient to account for the phenomena that they observed and experienced. This led them to the integration of the African traditional world view, also on the epistemic level, in their subsequent work as anthropologists. Through his reference, Bührmann relates her own work to theirs. In doing so, she claimed recognition and esteem for Bantu divination. (The importance of her 'politics of discourse', to connect her own explorations of Bantu divination, to the work of these scholars, has not been recognised by Landman, and the critics of Bührmann, such as Roper, that he refers to).

It may be acknowledged that, in spite of her long field work, she did not delve deeply into the metaphysical and religious aspects of Bantu divination.

With our theoretical perspective in mind, to investigate the resurgence of Bantu divination in South Africa as a 'reception', her pathways of access are of interest. She has been criticised for not delving deeply into the metaphysical and religious aspects of Bantu divination to accompany her extensive field work. She expressly stated that her interest lay with its ritual enactments and practices (ibid., p. 22). This she found a necessary complement to perceived deficits of Western culture (ibid., p. 22). As special elements of her interest in Bantu divination that converge with Jungian psychoanalysis, she enlisted the following:

"4. The ancestor concept of the Xhosa, especially as it is used . . . for the purpose of healing.

5. the Xhosa's attitude to dreams is that these are communications from the ancestors and may therefore not be ignored and that every effort must be made to understand the messages these dreams convey.

6. the roles of rituals, rites, ceremonies and sacrifices in the life of the . . . Xhosa.

7. the significance of the *intlombe* and *xhentsa* during which body and spirit find expression and are united in a beautiful and meaningful way." (Bührmann 1986, p. 17f.)

Bührmann thus outlines the common ground, which is the analysis of dreams, with the assumption that they have a 'transcendent function'—an aspect that Jung emphasised in his theory and practice (Jung 1984b, p. 259). Her presentation of Xhosa divinatory rituals is somewhat ambivalent. Instead of describing them in their own metaphysical and religious frame, with the meanings defined here, she reinterprets them in the frame of Jungian world view, based on the claim that they represent "archaic layers of the psyche, and the symbols from these still have power and meaning for them" (Bührmann 1986, p. 22). (This alludes to secularisation in the European context). Her discursive strategy integrates them into the European world view, but at the expense of ignoring some salient features. This may be observed for her report on the intlombe ceremonies, that she depicts thus:

> "At a good intlombe with the full participation of all those present a numinosity is engendered which stirs up archaic, long forgotten or ignored layers of our psychological and physical beings. These are experiences which lie outside our rational, logic and scientific way of being and functioning in this world. I could not ignore these mythic experiences because during my personal analysis I had fleeting glimpse of such happenings. It was however only after I had attended a two-day ceremony, which was devoted to practically uninterrupted dancing, I singing and talking, and which I tape-recorded that (…) I could discuss my experiential and intellectual understanding of the intlombe in terms of analytical psychology, in particular in terms of Jung's concept of the mandala a universal symbol of the collective unconscious. I now perceive the intlombe as a mandala in action." (ibid., p. 58)

It is remarkable that Bührman apparently did not care to enquire about the meaning of the ceremony in the frame of the Xhosa traditional world view (and philosophy), but declared it to be a manifestation of 'our archaic past'—thus ignoring the recognition of difference of world view—and as manifestations of a paranormal quality, reminding her of some of her own, that she vaguely redefines as 'mythic', thus brushing them away conceptually in order to present an interpretation of the ritual as 'mandala' in Jungian terms, as a "universal symbol" of a "collective unconscious". However, she does not explain in which features this should manifest itself. In terms of her 'discursive appropriation' of the Xhosa ceremony, she arrives at what interests her in particular, to perceive Xhosa rites as 'enactments' of Jungian concepts, and thus as meaningful complements, suited to be integrated into the repertoire of psychotherapy.

It is only then that she ventures into the realm of the 'other', to enquire about the meanings of the ritual elements that she observed and recorded. About the songs, she learns that these are inspired by the 'umbelini'. About these she learns:

> "Umbelini is a term for an important and sophisticated concept … The generic meaning is 'intestines' or 'gut' … generally used to describe a feeling of anxiety or anxious anticipation … experienced in the chest or abdominal area, with palpitations of the heart and a feeling of impending doom. The amagqira [diviners] ascribe a wider meaning to it and briefly call it 'life forces'. (…) One of the aims of the intlombe is to increase and raise the umbelini of the amagqira." (Bührmann 1986, p. 60)

Here, she finally arrives at a recognition of 'alterity', entering into the realm of the Xhosa world view as an ethnographic researcher. Yet, a certain tension is observable between an appropriating re-interpretation of the rite in Jungian terms, and recognition of the 'alterity' to learn about the meanings in their own context of the African world view and philosophy, in her description of details. It appears that this tension may have facilitated the reception of Xhosa divination to 'European' and modern readers, who could feel assured that these rites were meaningful in terms of Jungian depth psychology, and could thus be accepted. However, she tends to downplay the aspect of 'alterity'—which has rightly earned her critique—as evident in the following passage:

> "There is no doubt that the intlombe and xhentsa [slow ritual dance] evoke feelings and physical experiences which cannot be denied even though as yet there appears to be no entirely satisfactory explanation to account for these changes. Neurophysiological and biochemical substances are likely to play a role…" (Bührmann 1986)

With the vague phrases of 'undeniable inexplicable feelings and physical experiences', she vaguely acknowledges her experience of 'alterity', but is quick to attribute it to 'neurophysiological' factors, thus retreating to the 'safe ground' of Western rationalism, whose limitations she had criticised before.

Such reductionism is even adopted by the authors of a fine ethnographic study of the San ('Bushmen') world view, and its motifs, including practices of divination, that influenced the Xhosa. Thus, David Lews-Williams and David Pearce declared:

> "In the current climate of thought, spirituality has come to mean little more than 'other-worldliness', and the word is so heavily loaded with positive connotations, that any attempt to dissect it seems sacrilegious (. . .) We adopt a more materialist position. Spirituality cannot be understood without neurophysiology . . . Religion is not so much the attempt to explain the natural world . . . and to cope with death, as a way of coming to terms with the electrochemical functioning of the brain." (Lewis-Williams and Pearce 2004, p. xxiv).

What both leave unexplained is why the brains of the San should function so differently from those of people from the Middle East or of Europe, as to produce the distinct San mythology and religion. Their claim, that San art is essentially the same as the European cave art of the Ice Age, as an expression of a common 'shamanism', has been criticised for not taking regional differences of culture and religion into account (Ponomareva 2021).

David Lewis-Williams is an eminent scholar of the iconography of San rock art. He identified its symbolism, and promoted reading its images as representing a visionary world view, including states of trance (Lewis-Williams 2003).

The rationalistic approach, however, declared their cosmological aspects as illusionary, and, thus, their distinct phenomenological reality, warranted by San tradition—and similarly in Bantu tradition—as questionable. With such reattributions to postulated 'neurophysiological' factors, the cultural difference of this world view to that of European rationalism is disputed. The perceptions of Bantu and San diviners are condescendingly devaluated as 'hallucinations'—compared by him to states induced by opium, as Nicholas Conard explains (Conard 2006), as a source for the specific visions and cosmology of the San, in particular. David-Lews and Pearce claim that the tiered cosmos of many mythologies is the product of

> "neurologically generated sensations (. . .) Not only is this tiered cosmos produced by the human nervous system, it can also be verified by altered states of consciousness. (. . .) Passage between cosmological tiers is thus achieved by altered states of consciousness . . . that are in fact hallucinations." (Lewis-Williams and Pearce 2004).

The adage of 'religion as opium of the people' (Marx 1844) —formulated by Karl Marx in a different sense—is applied as an explanation for the visionary art of the San, as hallucinatory expressions of a universal pre-modern mind, without explaining why these culturally specific forms and motifs have evolved, as critically noted above. Furthermore, it ignores the claim that the visions of the San (and Bantu) diviners do indeed constitute perceptions of realms of reality that lie beyond what European modernity accepts as such. This violates an epistemic boundary that the eminent scholar on shamanism, Mircea Eliade, formulated succinctly, in view of its common motifs and cultural differences, when he declared that the images of shamanic flight that occur world-wide "cannot be fathomed exhaustively by psychological explanations; there remains an irreducible core, and this . . . may reveal us something about the true position of the human being in the cosmos." (Eliade 1975, p. 4). The enshrinement of the non-Western world view of ATR—with its explicit reference to 'spirits' as a source of knowledge (The Presidency 2004)—as basis for the practice of Bantu divination in South African law is, therefore, highly important to protect its epistemic foundations from a '(post)colonial' imposition of European naturalism, irreverent of 'alterity'. Although it may appear as a digression, the issue is critical to the understanding of Bantu divination. There is unanimous agreement that evidence of paranormal faculties of premonition, but also of clairvoyance, is indispensable for graduation as a Bantu diviner (Hall 1994). The encounter with spirits, in a state of trance, also belongs here (ibid., p. 100ff.). To dispute their experienced and observed reality

categorically as 'hallucinations' is—in the context of African culture—a gesture of callous, 'colonial' disregard, that denies the basis for epistemic dialogue and respect for 'alterity'.

The authors then ridicule the eminent role of dreams for the San as 'autistic'. Their art of dream interpretation—also important in Bantu divination—is likened to that of psychoanalysis, and devalued as 'outdated':

> "Today in the West, dreams are largely discounted as amusing or sometimes rather frightening, but essentially meaningless (apart from a couple of schools of psychology). (...) so too it is with the rest of the spectrum of consciousness. In the West people who talk much about their dreams do not earn respect. (...) In some societies, the autistic end of the spectrum ... is not only valued, it is also guarded." (Lewis-Williams and Pearce 2004, p. 34).

Given the profound influence of psychoanalysis and its dream interpretation (Freud 1900) that inaugurated the 20th century, as Sigmund Freud was aware of, on the sphere of psychology, with the key concept of the 'unconscious' and its expression in dreams; on the arts and literature of the 20th century, as in Surrealism or Phantastic Realism; on anthropology; and on general culture, which adopted concepts of psychoanalysis as household words, one may wonder which cultural environment the authors, apparently oblivious of this heritage, are referring to here as 'Western'. Considering the eminent role of dreams in the African traditional world view, and divination, these assertions come across as insensitive, also to their cultural context of writing. Their book on San mythology and symbolism fortunately exceeds their 'narco-materialist' reductionism, in its fine descriptive chapters, as on the San iconograph of rain (Lewis-Williams and Pearce 2004, p. 137). This is useful for the understanding of aspects of Bantu divination, as on 'rain' (Mbiti 1969, p. 174), making it a valuable source for some of the roots of Xhosa divination.

A contradiction is perceptible here between the apperception of a wider world view of African cultures, including 'transcendent' phenomena, and a rationalistic interpretation that eliminates or downplays—as 'mythical', 'archaic', or 'prerational'—some of their essential features, at the point where the encounter with the 'alterity' of another culture becomes unsettling. This is also observable in passages of Bührmann:

> "The intlombe, the ritual healing dance which creates as numinous atmosphere, confirms Neumann's statement. 'Originally all ritual was a dance, in which the whole corporeal psyche was literally set into motion.' (...) While I kept my critical conscious faculties in abeyance and for the time being just became an organ of reception, I did experience the above." (ibid., p. 66f.)

A tension appears here between the inner and outer phenomena, that she acknowledges to have experienced in the 'Xhosa realm', and their reductionist, generalising reinterpretation that evokes a German saying: 'Wash my fur, but don't make me wet". This was unlike Laubscher, who sought to accept the phenomena which he observed and experienced with concepts of European tradition, thus venturing further into the realm of Xhosa 'alterity' than Bührman.

Her position, however, as founding member of the Jungian association of South Africa, together with the influential author Laurens van der Post (Landman 2019, p. 137ff.)—who wrote about the San (Bushmen) (Van der Post 1962), promoting an image of the San as living a spirituality of connectedness to the cosmos (Lewis-Williams and Pearce 2004, p. xxvii), that resembled Bührmann's views—has certainly contributed to her success in promoting interest in Bantu divination in Jungian analysts world-wide, in the context of the emerging trans-cultural psychology (Landman 2019, p. 110ff.). This legacy remains alive.

Bührmann's connection with the literary author Laurens van der Post, by their common interest in the divination and spirituality of the indigenous people of South Africa, may be related to the reception of this heritage in South Africa that had been going on for a century by then. In the Cape Province, and in Namibia, the language and mythology of the San (formerly known as the 'Bushmen') and the Khoi (formerly called, somewhat derogatorily, 'Hottentots') were studied eminently by the German scholar Wilhelm Bleek

(Bleek and Lloyd 1911) and received by literary authors of the Afrikaans language. This reception, also from other collectors, especially Gideon R. Von Wielligh (Von Wielligh 1919–1921), created archives of mythology, from which literary authors of the emergent Afrikaans language drew in their endeavour to be rooted in the mythology and motifs of their land, especially in the south-western part of South Africa.

This reception is in the focus of Helize Van Vuuren, who investigated it with a differentiated set of literary hermeneutical tools, for several San myths, e.g., "Night and Darkness and their three Daughters" (Van Vuuren 2016, p. 4ff.). The ongoing influence of this reception, as for the work of the eminent 'poeta doctus' of Afrikaans literature of classical modernity, Nicolaas P. van Wyk Louw, who grew up in a zone of cultural contact with the San descendants, is included in her present research on him. The striving to combine the vast heritage of European philosophy and literary traditions, as well as issues of spirituality, and of modern poetics, including the symbolic and the mythological, is connected by him with attention to the symbolism and myths of the San culture of his native land. It is part of a movement of reception of San myths into Afrikaans poetry from the early 20th century on. This context is important as background of the engagement of Laubscher, Bührmann, van der Post, Brink, and other cultured Afrikaans professionals, to encounter, experience, and receive indigenous divination of the Bantu and the San. It is a broad movement of 'incorporation of alterity', to find the own 'voice' in their 'land' that is 'inscribed' with their myths, based on their mediumistic perceptions. Thus, van Vuuren writes:

> "The literary aspects of van Wielligh's version of the/Xam archive are noticeable for their poetic sparks they lit in many instances, and from which Afrikaans poetry grew in the early 20th century. Jan FE Cilliers' (1908) poem, '*Die vlakte'* ('The plains') seems intimately bound through its title to the/Xam myth of the second daughter of 'Ga and 'Gagen. (...) This poem describes the plains as a sleeping woman, with all of its "life held in her bosom". (...) Similarly with Eugène Marais' 'The Dance of the Rain' ('*Die dans van die reën'*, 1921) and the/Xam myth of !Khwa. (...) The rain, like the plains, is one of the sisters as envisioned in/Xam mythology." (ibid., p. 84f.)

Thus, the San gives voice to early Afrikaans poets, and mythological meaning to their experience of the land. Interestingly, the banishment of indigenous divination is also a motif in one of the poems:

> "Marais' *Dwaalstories* (Marais 1927) ('Wandering Tales' or 'Tales of Trickery')...
> contains a further handful of ... memorable poems, all from the hunter-gatherer perspective, such as 'The Sorceress' ('*Die towenares'*) that describes an old medicine woman or sorceress who has been chased out of her clan along with her two granddaughters and banished to live alone...
>
> 'What becomes of the girl who is always alone?
>
> She no longer waits for the hunters to return (...)
>
> She no longer hears the dancing song; -
>
> The voice of the storyteller is dead.
>
> No one calls to her from afar
>
> To talk sweet words (...).'" (Van Vuuren 2016, p. 85)

This figure is to be read as an allegory for the banishment of indigenous divination, and for the mutual loss that ensues. It captures the loneliness and the mourning of the tellers of myth—symbolising the death of San culture—and that the sorceress/diviner can no longer hear the rain-bearing wind, nor the dancing song to call the rain. It also tells of the misery of the diviner, upon whom no one calls any more, to talk 'sweet' life-giving words. Marais had a deep sense of the loss of a 'holistic' life-world encoded in San myth, and its divination, that would leave the white colonists with a bleak and meaningless land.

The eminent poet of Afrikaans modernity, Diederik Johannes Oppermann, who was born in Zululand, adopts the figure of the 'sangoma', to depict himself as 'poeta vates' ('mantic poet' or 'seer'), as Helize van Vuuren tells:

> "in Dolosse ['oracle bones'], the poet features as a . . . 'sangoma', who performs acts of divination by throwing the bones, so as to discern the future. (. . .) What he foresaw for South Africa . . . was 'a cold spiritual hell'. The poet's reading of the bones . . . produced poetry of the highest intensity, although his own catabasis or 'descent into hell' started here." (ibid., p. 183)

Out of this background of poetic retrieval of indigenous divination from the early 20th century on, as a necessity to live meaningfully in the land, the passionate endeavours of Laubscher, Bührmann, and others are to be understood: to encounter the 'other' for the sake of the 'self'. Their work paved the way for the appreciation and retrieval of indigenous divination by black African scholars and diviners such as Mlisa. It also echoes its retrieval, at the same time, in the 1920's, by the black Sotho poet Thomas Mofolo in his historical novel on the founder of the Zulu Empire, king Chaka (1787–1828), in which Bantu divination is the guiding force, presaging the king's rise and fall, and endowing him with guidance, protection, empowerment, and mentorship (Mofolo 1981). (A quote from this novel biography may convey the spirit of this realm). Mofolo describes the encounter with the isanusi, who emerges as his spiritual mentor. Chaka interrogates the stranger, whom he recognises as a diviner, at their first meeting, as follows:

> "When the doctor finished those words . . . Chaka once more asked . . . and the doctor then said: 'As a human being, you might think that I heard of your affairs through someone else, yet it is not so. But, in order that you shall believe in me, I shall tell you one small matter which is known to you alone. In your tuft of hair there is a medicine to bring you luck and kingship. When you adjusted your blanket I saw, I who have the power of vision, and I became aware . . . that you were once visited by the great master who comes from those who have departed [i.e., the ancestral spirits], who are above, and that master was highly pleased with you. Besides, my eyes see things which have already passed, can see that you were frightened when the master was with you, so much that your hand refused to leave this tuft of hair, in the manner you had been instructed by the woman doctor who is now gone.'" (Mofolo 1981, p. 38f.)

This exploration of Bantu divination and worldview is comparable, in some ways, to the dedication of literary authors of Modernity in Brazil to the mythology of African (Amado 1988) and Amerindian people, in the striving for a distinctly Brazilian literature of modernity (De Andrade 1928) that integrates the African world view in a poetics of Magic Realism. Several Afrikaans authors drew motifs from this source—André Brink in a similar way as Amado (Brink 2005).

This literary reception mirrors what took place in the Jungian exploration and reception of Bantu divination, and its underlying world view. A systemic 'syncretism' emerged, both in the sphere of literature as in the field of psychoanalysis, into which concepts and practices of this source were integrated, in a distinctly South African synthesis, comparable to the Brazilian, that avoids the negation of 'alterity' by rationalistic reductionism, but dares to venture into the world view, and perceptions of inner and outer phenomena of the African traditional world view, expressed eminently in Bantu divination.

### 5.3. Lily Rose Nomfundo Mlisa (Academic Psychologist, Igqirha)

The transition of the reception of Bantu divination from white professionals in the fields of psychiatry and psychoanalysis to black African professionals as protagonists, may be shown with Lily Rose Nomfundo Mlisa, an academic psychologist at the University of Fort Hare. Following the call of latent mediumistic endowment since her childhood, she underwent training and initiation as an igqirha, a Xhosa diviner, in the frame of her traditional culture and its institutions, in a structured long course of training and

examinations. She wrote her dissertation in anthropology on this subject (Mlisa 2009). Her dissertation is rare for its detail, and its combination of the academic description with her personal experience. The report of James Hall (1994) is likewise rich in detail, adding facets. These presentations are complemented by academic research, such as that of John Janzen (1992). It is an incipient process of mutual elucidations of '-emic and -etic' theories, observable in several publications of academics initiated or engaged in Bantu divination (Wreford 2008).

Since then, Mlisa lectured on Bantu divination at international academic conferences, such as in Juiz de Fora (Minas Gerais, Brazil) in 2018 (Lages 2018), with a presentation on her personal journey to become a diviner. Her title: "The Spectre of the 'Other'" refers to the 'alterity' of the African traditional world view and its divination, perceived as a potentially dangerous shadow from the past, and a challenge to an academic, such as herself, in her retrieval of this heritage, that she felt to be an inner necessity. She also lectured, and held seminars at conferences of the Jungian analytic society, both in South Africa, repeatedly in Cape Town (Mlisa 2017), and at the international conference of Jungian analysts in Vienna in 2019. Of the latter, she reported:

> "Presentations like the one I presented recently at the IAAP international Jungian conference in Vienna, (https://iaap.org; https://bit.ly/iaap-cloud, accessed on 28 February 2024, International Association for Analytical Psychology 2019) provides another evidence. We were four presenters and I was the third to present. Immediately after my presentation a conference hall with more than 1000 (no exagerations, exhibits a standing ovation as expression of their passion, reception and another spiritual intuitive linkage) participants suddenly stands up clapping hands, row by row and all were up. Others calling "hallelujah; Amen!!" others crying. The time for the next presenters was delayed by the emotional moment and the chair person tried to calm the atmosphere. I had a range of individual sessions after the plenary session, and right through until I left. It was good that our plenary was on the last day of the conference. The word spread to the extent that after 3 days, I had to present the same presentation at a Jungian Institute, in Zurich. It was evident, they were waiting for me. Did I meet different questions from the ones in Brazil and South Africa, no!" (Mlisa 2020)

The emotional impact of her persentations is important to note because it shows the spirit, interest, and appreciation in her presentation and in Bantu divination by these professional psychoanalysts, and academics. (The emotions being important for the further reception). This was also tangible in Juiz de Fora, where some priestesses, priests, and diviners of Umbanda were among the academic participants. The similarities were recognised, and some interesting differences that appeared to be mutually enriching and remain to be explored.

A visit to a temple of Candomblé in Rio de Janeiro after the conference emphasised the religious dimension of Bantu divination. An event here is meaingnful to report:

> On the temple ground, we were shown some jackfruit trees, with their large fruit growing from the trunk. We were told that these trees, not indigenous to Brazil, were regarded as 'ancestors' trees' in Candomblé. Nomfundo Mlisa stood frozen, as if thunderstruck. Then she told: 'I saw this tree in a dream, four years ago. I have never seen such a tree in my life before." (Witnessed by myself, in situ. 27 October 2018).

The significance of 'ancestors' in Bantu cosmology is that they act as guardians and mentors of the living, especially of their descendants from the realm of the afterlife (Mbiti 1991, p. 75f); furthermore, they may act as divinely inspired messengers (ibid., p. 227). They tend to appear in dreams. The deep emotional resonance that her presaging dream, and its realisation here and now on this temple ground evoked among all can well be imagined. The phenomenal world of Bantu divination and world view had become experiential reality in this manifestation.

The process to become a diviner is a well-structured professional course, as Mlisa declares:

> "It is critical to note that it is a training like any other professional training with entry level requirements, pedagogical instructions, criteria for assessment and condonation including graduation and internship after graduation. The challenge is, its orality, a challenge that is being managed currently. (. . .) All aspects of life as World Health Organization (WHO), (1995) specifies are always included in the therapeutic regime such as: emotional and psychological, spiritual, physical, and social aspects." (Mlisa 2020, p. 222f)

This, she states, helps its integration with therapeutic approaches from modern psychology. She describes its stages thus:

> "The training evolves through seven stages including sub-stages and diffusion of certain stages at certain occasions. (. . .) First stage: Prediction stage of a chosen Igqirha. The person is chosen as a healer by her ancestors at conception (. . .) Experiential narratives of amagqirha reveal that indications that a person has ubizo, (the calling) to thwasa can be identified as early as at birth. Second stage: manifestations of sins to indicate the ubizo (calling): The first stage signs persist and often are now mixed with some sicknesses that may not be treated successfully with any allopathic treatments. Various illnesses and perhaps seeing, shades, hearing voices and very alarming dreams, others see snakes, . . . clanship animal totems, and many others." (Ibid., p. 227)

About this critical stage, she comments, that those 'afflicted' are often misdiagnosed as mentally disturbed, and treated for delusion disorders—although, by tradition, the resolution of this crisis follows in the subsequent training, and not by psychiatric means. The symptoms may overlap, but the aitiology, and resolution is different:

> "This is often diagnosed as pathology by western trained doctors and priests at church; and at times ultimately attracts unnecessary admission in a mental hospital." (ibid., p. 227)

The differential diagnosis to ascertain if a mediumistic calling is present or a mental disturbance is made by professionals:

> ". . .to verify or confirm what is happening. It is well known tradition to resemble ukuthwasa with sickness as it is often revealed through a series of syndrome signs and symptoms . . . or crises. Hence, amaXhosa refer it to ingulo emhlophe (white sickness)". (Mlisa 2020, p. 227)

> If this is established then actual training can begin, often with challenging demands. To reject it is also said to come at a cost, of difficulty to find the ealier equilbrium. (Wreford 2008, p. 10f.)

> "Third stage . . . of intense afflictions, . . . crises and sickness intensify and a family is forced to do something. (. . .) Verification and confirmation of a need to start the journey of ukuthwasa becomes more apparent and in most becomes a last straw. After confirmation, the training often starts immediately or later on depending on the socio-economic status of the family, as the processes is expensive. No church prayers or doctors' treatment brings a cure, except cultural training. It is the sickness of the ancestors a pre-requisite to be trained as igqirha." (Mlisa 2020, p. 228)

> At this point, the formal training begins when a suitable mentor is found—who is often revealed, mutually, through dreams. Thobeka Wreford reports an account of a trainee, Nosibele, about her 'finding process', after a long crisis:

> "'Then, in 1968 I had a dream which led me to my first teacher. Unfortunately, in the dream I did not see his name, but only his attire. . . The dream told me that I must travel to Zimbabwe to find him, to Myadiri, the place where my sister's husband came from.'" (Wreford 2008, p. 104)

Having arrived at the place after a long journey from Cape Town, she is told that one needed to know the name of the diviner with whom she wanted to train, or else the search would be futile. In a following dream two names are revealed:

> "'And I dreamt of two names: . . . Masvingi . . . the second . . . was Kangai. In the morning the nephew came . . . I told him the names, and he said the first one had died long ago, but the second one, 'It's his son . . . and he lives three miles away. I walked with him to find this man (. . .) . . . like the Shona do we waited outside and called and knocked politely at the door, and when it opened the man in my dreams stood there and he looked shocked, and then he said, 'Where have you been all this time? I have been waiting for you.' Then I stayed there . . . Kangai had told me . . . I must not hurry it. I must wait for the ancestors to tell me.'" ([Wreford 2008](#), p. 105)

Such guiding and predictive dreams are a regular feature, and confirm the validity of the calling. This feature also shows up clearly in Mlisa's dream of the jackfruit tree, that she saw four years later in the Candomblé temple ground in Rio de Janeiro (told here above). The next stages were as follows:

> "Fourth stage. Confusion, resistance and/or acceptance. This stage also forms extended part of the screening process as means of verifying the presence of the calling. (. . .)

> Fifth stage: ukuvuma ukufa: acceptance of the calling. The actual training begins. (. . .) with two ritual activities: cleansing and acceptance during each sub-stage. (. . .) Verification has been finalized and a prospective initiate is ready to begin a long cultural spiritual journey.

> Most respondents questioned me both in Brazil and Vienna Jungian Conference about why the training takes so long (5 years and more) and what happens during the training and how does one cope. (. . .) Common in all these substages are: performance of variety of cultural rituals for cleansing, purification, intensification of ties between ancestors and the initiate, restrictions to be observed and disciplinary and ethical conditions to be observed by the initiate. (. . .) Sixth stage: Ukuphuma. This is the last stage of the intensive training that prepares umkhwethato be a fully fledged healer. It proceeds to graduation. It consists of four ritual activities (. . .) The seventh stage: Ukuphinda indlela . . . or internship stage." ([Mlisa 2020](#), p. 229)

This detailed account gives a brief insight into the process leading to training, as an answer to a call, and into aspects of the structured training. It appears as necessary to report the outlines here, to give the reader unfamiliar with this field an idea of what Bantu divination and the process to become a diviner comprise. For the audiences of academic scholars at the conference at Juiz de Fora, and for Jungian psychoanalysts, both with their long years of intensive training and formation, this 'discourse', that the training process of Xhosa diviners takes five years, in a structured process, with demanding examinations, certainly rings well. It puts them on par, in terms of experience, formation of personality, and expertise, thus indicating a good basis for dialogue and respect, also in the frame of the beginning academic engagement of the profession. Mlisa indicates the pathway with reference to other scholars, combining engagement academic accomplishment with practice to Bantu divination:

> "I fully agree with . . . [Van Binsbergen](#) ([2003](#)) (. . .) and [Masoga](#) ([2001](#)), a fully-fledged trained traditional healer, academic and scholar and a professor, dean of a faculty, that amagqirha divination system is dynamic, because Africans shower new problems and options with fresh meaning, firmly tying emergent orders into the previous ones." ([Mlisa 2020](#), p. 236)

Being an accomplished academic psychologist, anthropologist, and with strong ties to the international community of Jungian analysts, Mlisa's emphatic conclusion shows a

pathway, in which respect and recognition for Bantu divination, and the world view on which it is based, are demanded.

## 6. Conclusions

A most important finding emerging from the literature on divination is that the 'religious' and the 'therapeutic' spheres are connected in Bantu divination, with its diagnostics and its therapies. They are regarded as essentially connected. With the legal recognition of Bantu divination, this is acknowledged, endorsed, and enshrined in law. This fact is highly significant in the context of global modernity, in which issues of physiological and psychological health are chiefly defined in secular paradigms. (The incipient recognition of the healing effets of spirituality, are often regarded merely as an imaginative effect, or as a basis for sound practices of meditation). With the legal recognition of Bantu divination, a case is made for a 'distinct 'South African Modernity' that understands the relation between the spheres of medicine, psychology, and spirituality, differently from secular paradigms in the 'West'.

In this essay, I showed how the recognition of Bantu divination in the sphere of psychotherapy in South Africa was facilitated and promoted by psychiatrists and psychoanalysts of Jungian orientation, enhanced by a cultural movement of dedicated studies of indigenous divination, mythology, and African world views on which they are based. I also looked at contempory strivings in literature and poetics, to integrate them into the Afrikaans (and later Anglophone) cultures of South Africa. It emerged that these movements facilitated the retrieval of Bantu divination by black African academics, of professionals in the field, and scholars beyond.

The parallel process of reception of Bantu Divination in 'African Initiated Churches' is to be kept in mind. Its presentation, however, lays beyond the frame of this essay. (It contributed powerfully to the revival of Bantu divination—redefined as the charism of 'prophecy'—in the frame of Christian churches, in this predominantly Christian country). This is to be appreciated as a powerful factor, reinforcing its presence in culture and society.

The reception of Bantu divination and its world view was strongly based on the metaphysics of C.G. Jung's concept of the human individual and collective soul, and its heritage. In it, the traditions of divination, as by dreams, and the philosophies, especially of Neoplatonism and of Hermeticism, which Jung studied and integrated, have a crucial role. They provide a bridge to the transcendent world view of Bantu cosmology, in which (ancestral) spirits and precognition have a firm role. The integration of spiritual aspects is likewise a feature of both traditions. In this essay, some lines of tradition and resonating motifs have been traced.

In this essay, I showed how Jungian psychoanalysis became the key factor for the reception of Bantu divination, by providing a conceptual frame in which the resonance between African and European traditions of divination could be received by academic and professional psychoanalysts. The resulting legal recognition secures this reception and emergence in modern South African society. The call for further integration of Bantu divination with academic scholarship, expressed in the law, indicates that this process is not complete. Limitations of Jungian analytic psychology, regarding the spiritual world view, and religious aspects of Bantu divination (of which Laubscher was more mindful than Bührmann) have been noted here. They require further investigation, as to the philosophical traditions on which both are based. In this study, it emerged that this reception proceeded in contexts of developments in literature, and in religion, in African Initiated Christianity, Esotericism, and Afrobrazilian Umbanda. This outlines a wide field for explorations in the fields of psychotherapy, religion, mediumism, literature, and history of philosophy. In particular, the history of European philosophy of divination, the forms of practice, of structured training in modernity, and the poetic reception, appear as realms for fruitful further studies. In this essay, in which I traced an interesting process of intercultural reception in South Africa, and the ongoing developments resulting from it, interconnections between these spheres became apparent that also merit a view across the Atlantic.

**Funding:** This article received no external funding.

**Institutional Review Board Statement:** Not applicable.

**Informed Consent Statement:** Not applicable.

**Data Availability Statement:** No new data were created or analyzed in this study. Data sharing is not applicable to this article.

**Conflicts of Interest:** The author declares no conflict of interest.

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
