# Peer review of "The Reception of Bantu Divination in Modern South Africa: African Traditional Worldview in Interaction with European Thought"

_religions, doi:10.3390/rel15040493_

Round 1
Reviewer 1 Report
Comments and Suggestions for Authors
The article is very well researched, is intriguing, and original. It brings together many strands of thought: African religions, Jung, esotericism, modern acceptance of divination. Could the title be shortened in some way? It is cumbersome. Perhaps the article tries to do too much by bringing in Brazil. I'm not suggesting that this portion should be omitted, but something worth considering. My major comment will have to do with style and English below.
Comments on the Quality of English LanguageIt needs to have someone other than the author read it for the flow of English. There is no problem with the English but needs editing. I say other than the author since it is more difficult for an author to detect this items. Following are some examples, but only some. The author might disagree on some point of style.
- line 77 - "of" Bantu
- use of commas -- lines123-24, 145, 152 -- usually commas not necessary that disrupts the flow.
- line 135 --"a" theory -- an example of need for careful editing
- lines 140-41 -- allow to speak
- line 148 - elements
- at times there is a change of font -- e.g., line 288, 1089
- line 346 -- word 'that' missing --
- line 389 --'the'
- line 392 -- co-founder
- line 538 - no 'the'
- line 658 - cap for Nelson
- line 1137 - spelling for 'significant', 1141 'understands
These are representative examples. I did not record all of those that need editing. Editing will make it more readable and less burdensome for the reader, The topic and research deserve this.
-
Author Response
Dear Reviewer,
please find my response to your comments in the (response to reviewer 1" document attached, and the changes in the text, that I made, in the attached revised version of my essay, which also includes my revisions in response to the comments by the other reviewer, and the corrections that I made after language and style editing by a retired professor of English, later of Comparative Literature, as well as additions in response and for further clarifications.

Reviewer 2 Report
Comments and Suggestions for Authors
Please, see the highlighted parts and attend to them.
the argument of the demise of ATR seems contradictory to the reality of syncretism. i suppose that the word 'demise' is too strong because the subsequent arguments do not suggest that ATR is dead.
Towards the end of the article, long quotations appear to weaken the strength of the paper and the capacity of the author to capture the arguments of the authorities being reviewed. it is suggested that some of them should be rendered in prose.

Comments on the Quality of English Languagethe language is good, except for a few, which have been highlighted in the attached.
Author Response
Dear Reviewer,
thank you for your comments!
In response, I marked the passages that you commented in blue, with the changes in red script.
I shortened the title. (I also shortened a long quote.)
I reworked the conclusion.
I also took up the suggestion for further clarification by adding some passages. These are in dark red script. I asked a friend, a retired professor of English, later of Comparative Literature, knowledgable on this subject matter, for a proofreading of style and language. the changes which I made accordingly are in brown script.
I adjoin two documents: my respnse to our review, and the reworked version of the essay.
(My responses to the other reviewer are highlighted in yellow.)
best regards!
